



# A vortex-based tip/smearing correction for the actuator line

Alexander Raul Meyer Forsting[1], Georg Raimund Pirrung[1], and Néstor Ramos-García [1]

[1]DTU Wind Energy, Technical University of Denmark, Frederiksborgvej 399, 4000 Roskilde, Denmark

**Correspondence:** Alexander R Meyer Forsting (alrf@dtu.dk)

**Abstract.** The actuator line was intended as a lifting line technique for CFD applications. In this paper we proof - theoretically and practically - that smearing the forces of the actuator line in the flow domain necessarily leads to smeared velocity fields. By combining a near-wake representation of the trailed vorticity with a viscous vortex core model, the missing induction from the smeared velocity is recovered. This novel dynamic smearing correction is verified for basic wing test cases and rotor simulations of a multi-MW turbine. The latter cover the entire operational wind speed range as well as yaw, strong turbulence and pitch step cases. The correction is validated with lifting line simulations with and without viscous core, that are representative of an actuator line without and with smearing correction, respectively. The dynamic smearing correction makes the actuator line effectively act as a lifitng line, as it was originally intended.

## 1 Introduction

The actuator line (AL) technique developed by Sørensen and Shen (2002) is a lifting-line (LL) representation of the wind turbine rotor suitable for computational fluid dynamics (CFD) simulations. It captures transient physical features like shed and trailed vorticity (including root/tip vortices) , without the computational cost associated with resolving the full rotor geometry. The AL model thus enables Large-eddy simulations (LES) of large wind farms in realistic, turbulent atmospheric boundary layers (Vollmer et al., 2017).

However, different to LL vortex formulations the blade forces are dispersed in the flow domain - most commonly in form of a Gaussian projection - to avoid numerical instability. A length scale - also referred to as smearing factor - controls this force redistribution, whose lower limit is linked to the grid size through numerical stability requirements (Troldborg et al., 2009). Mikkelsen (2003) observed a large sensitivity of the blade velocities to this length scale, which consequently also propagated to the blade forces. Especially in regions along the blade exhibiting stark load changes, as around the root and tip, forces are substantially over-predicted. Meaning this effect is exacerbated by non-tapered and low aspect ratio blades. As actuator disc formulations suffer from similar issues towards the blade tip, their Glauert (1935) type tip corrections are also frequently applied to ALs (Shen et al., 2005). Yet, these correct discs for missing discrete blades and thus should be unnecessary - strictly even invalid - for ALs. Shives and Crawford (2013) and Jha et al. (2014) achieved a reduction in the force over-prediction





by varying the originally fixed smearing factor with respect to the blade chord. Yet, their methods cannot decouple the blade forces from the smearing length scale: a smeared force distribution in the flow domain necessarily leads to lower induction at the blade - increasing lift and drag - compared to an actual LL with a concentrated, spatially singular force.

## 1.1 The vortex smearing hypothesis

Shives and Crawford (2013) noticed the similarity between the velocities induced across an actuator line and those predicted by a viscous vortex core model. These models include the limiting effect of viscous shear forces on the induced velocities around vortex cores. A similar comparison of the swirl velocities about an infinite vortex line are shown in Fig. 1 - here with a Lamb-Oseen vortex core model (Lamb, 1932; Oseen, 1911). Without viscosity (inviscid) the velocities approach infinity towards the vortex centre. The startling agreement between the Lamb-Oseen and AL velocities was first demonstrated by Dag
et al. (2017). The Gaussian body force smearing in the AL technique thus produces similar swirl velocities as a viscous vortex. As the AL should in principle induce the same velocities as a LL - equivalent to the inviscid solution - the missing induced velocity in the AL model (marked area in Fig. 1) can be approximated following Dag et al. (2017) by:

$$\Delta v_\theta(r) = \overbrace{\frac{\Gamma}{2\pi r}}^{\text{inviscid}} - \overbrace{\frac{\Gamma}{2\pi r}\left[1 - \exp\left(-r^2/\epsilon^2\right)\right]}^{\text{viscous core}} = \frac{\Gamma}{2\pi r}\exp\left(-r^2/\epsilon^2\right) \tag{1}$$

Here $\Gamma$ represents the vortex line's circulation, $r$ is the distance from the vortex core and $\epsilon$ the length scale used in the force
smearing. This formulation can be split into an inviscid and viscous/smearing contribution:

$$\Delta v_\theta(r) = \overbrace{v_\theta(r)}^{\text{inviscid}} \underbrace{f_\epsilon(r_\epsilon)}_{\text{smearing}} \quad \text{with} \quad v_\theta(r) = \frac{\Gamma}{2\pi r}, \quad f_\epsilon(r_\epsilon) = \exp(-r_\epsilon^2), \quad r_\epsilon = \frac{r}{\epsilon} \tag{2}$$

If this viscous behaviour of the force smearing in AL simulations would be limited to the bound vortex representing the blade, then it would not influence the blade forces as long as the blade is straight. Yet Dag et al. (2017) argued the trailing vortices (in the wake) to exhibit same viscous core, as they originate from the bound vortex. Hence, the wake of an AL is inducing lower
velocities at the blade than in case of a LL. The missing velocity can be estimated from the viscous core equivalence and thus correct the velocities at the blade. This mostly impacts blade forces by changing the angle-of-attack at the blade sections.

## 1.2 Contributions of this paper

Dag et al. (2017) corrected AL simulations of a rectangular wing and two rotors with different aspect ratios by recuperating the missing induced velocity introduced by the viscous core. For all their simulations they were able to show the beneficial
effect of the correction on the blade load distribution - represented by more physical behaviour, especially towards the tip and root. However, their implementation of the correction did not fully couple the flow-field with the blade forces and the induction correction.

The major contributions of this paper are:

- The development of an iterative, dynamic and numerically stable smearing correction.





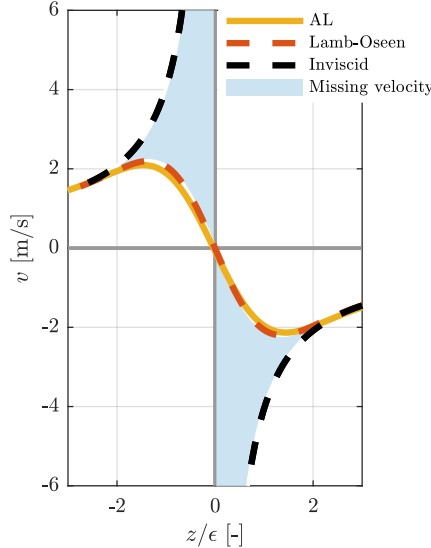

**Figure 1.** Distribution of the vertical velocity component in a plane orthogonal to an infinite vortex line (along $x$) obtained from either an inviscid or viscous (Lamb-Oseen) theoretical vortex and an actuator line CFD simulation.

- – A theoretical proof of the force smearing - vortex core equivalence.

- – Proof of the vortex core inheritance in trailed vorticity.

- – The confirmation of the missing velocity assumption by comparing LL simulations with/without viscous core and AL results with/without correction.

5  The test cases include constantly and elliptically loaded wings as well as rotor simulations of a multi-MW turbine covering the entire operational wind speed range. The correct dynamic behaviour of the new correction is verified through yawed inflow and pitch step simulations.

## 2  Proof of force smearing - vortex core equivalence

The equivalence between the velocity field induced by an AL and a viscous vortex can be derived directly from the incompress-
10  ible Navier-Stokes (N-S) equations. This proof follows the approach by Forsythe et al. (2015) that successfully connected an AL's vorticity field to its force projection. Starting by taking the curl of the incompressible momentum equation, the vorticity transport equation is obtained ($\boldsymbol{\omega} = \nabla \times \boldsymbol{u}$)

$$\underbrace{\frac{\partial \boldsymbol{\omega}}{\partial t}}_{\substack{=0 \\ \text{steady}}} + (\boldsymbol{u}\nabla)\boldsymbol{\omega} = \underbrace{(\boldsymbol{\omega}\nabla)\boldsymbol{u}}_{\substack{=0 \\ 2D}} + \underbrace{\nu\nabla^2\boldsymbol{\omega}}_{\substack{=0 \\ \text{inviscid}}} + \nabla\frac{\boldsymbol{f}}{\rho} \tag{3}$$



Here $\nu$ is the viscosity, $\rho$ density and $\boldsymbol{f}$ the body forces from the AL. The flow around a blade is nearly two-dimensional, as the span-wise flow is negligible. Without modelling the blade boundary layer also viscous effects can be disregarded. Furthermore the relationship between body force and flow-field is quasi-steady, as they balance faster than the flow evolves. Cancelling the respective terms and noting that in 2D flow ($y - z$ plane) $\boldsymbol{\omega} = \omega_x \hat{\boldsymbol{e}}_x$ and $\nabla = (0, \frac{\partial}{\partial y}, \frac{\partial}{\partial z})$:

$$(\boldsymbol{u}\nabla)\omega_x \hat{\boldsymbol{e}}_x = \nabla \frac{\boldsymbol{f}}{\rho} \tag{4}$$

Assuming the drag to be negligible, the force - in the form of lift - exerted by the AL on the flow in terms of its circulation $\Gamma$ becomes

$$\boldsymbol{f} = -\boldsymbol{f}_{\text{aero}} g(r) \tag{5}$$

$$\boldsymbol{f}_{\text{aero}} = L = \rho \boldsymbol{u} \times \Gamma \hat{\boldsymbol{e}}_x \qquad g(r) = \frac{1}{\pi \epsilon^2} \exp(-r^2/\epsilon^2) \tag{6}$$

Here $g$ represents a 2-D Gaussian force projection with $r$ indicating the distance from the AL. Inserting these expression into Eq. (4) and exploiting standard matrix transformation and mass conservation[1]

$$(\boldsymbol{u}\nabla)\omega_x \hat{\boldsymbol{e}}_x = \nabla(\Gamma g \hat{\boldsymbol{e}}_x \times \boldsymbol{u}) = (\boldsymbol{u}\nabla)\Gamma g \hat{\boldsymbol{e}}_x \tag{7}$$

$$(\boldsymbol{u}\nabla)\omega_x = (\boldsymbol{u}\nabla)\Gamma g \tag{8}$$

Due to mass conservation the $\boldsymbol{u}\nabla$ term can be inverted thus giving a direct relationship between the force projection and vorticity:

$$\omega_x = \Gamma g = \frac{\Gamma}{\pi \epsilon^2} \exp(-r^2/\epsilon^2) \tag{9}$$

As the body force is axially symmetric, the vorticity only induces tangential velocities

$$\omega_x(r) = \frac{1}{r}\left(\frac{\partial r u_\theta}{\partial r} - \underbrace{\frac{\partial u_r}{\partial \theta}}_{\substack{=0 \\ \text{axisymmetry}}}\right) \Rightarrow u_\theta = \frac{1}{r} \int_0^r r\omega_x(r) dr \tag{10}$$

Inserting Eq. (9) and integrating gives the swirl velocity induced by a smeared body force

$$u_\theta = \frac{\Gamma}{2\pi r}\left[1 - \exp(-r^2/\epsilon^2)\right] \tag{11}$$

This expression equals that of the Lamb-Oseen vortex, only with the viscous core radius replaced by the smearing coefficient[2]. This marks the theoretical confirmation of the observations by Dag et al. (2017), which additionally indicates that a viscous core behaviour with an AL requires inviscid, two-dimensional and locally steady flow conditions.

---

[1] $\nabla \times (\hat{\boldsymbol{e}}_x \times \boldsymbol{u}) = (\boldsymbol{u}\nabla)\hat{\boldsymbol{e}}_x - (\hat{\boldsymbol{e}}_x \nabla)\boldsymbol{u} + \hat{\boldsymbol{e}}_x(\nabla \boldsymbol{u}) - \boldsymbol{u}(\nabla \hat{\boldsymbol{e}}_x) = (\boldsymbol{u}\nabla)\hat{\boldsymbol{e}}_x + 0 + 0 + 0$
[2] Note that in the $x$-$y$ plane the circulation would be $-\Gamma$





## 3 Numerical methodology

### 3.1 Actuator-line simulations

The discretised incompressible Navier-Stokes equations are solved with DTU's CFD code EllipSys3D (Sørensen, 1995; Michelsen, 1994a, b). Details on the numerical techniques are given in Meyer Forsting et al. (2017). For all comparisons

with the LL code, the RANS equations are solved using the $k$-$\omega$ shear-stress transport turbulence closure of Menter (1993). Only the turbulent inflow cases in Sect. 5.2.4 are computed with the DES technique of Strelets (2001).

The AL model was implemented by Mikkelsen (2003) in EllipSys3D. We employ a version utilising three-dimensional Gaussian force projection, which follows the original formulation of Sørensen and Shen (2002). The smearing length scale is twice the grid size as recommended by Troldborg et al. (2009) to guarantee numerical stability. To ensure the blade tip to

10 remain inside a single cell during one time step $\Delta t < \Delta x / (\Omega R)$ - with mesh spacing $\Delta x$, rotor radius $R$ and rotational speed $\Omega$. Without rotation the term $\Omega R$ is replaced by the advection speed of the wake.

The numerical domain for the rotor simulations is discretised in a verified, standard manner (Meyer Forsting et al., 2017; Troldborg et al., 2009). It consists of a box with $25R$ side length that contains an inner box with a uniformly spaced refined mesh of $3.2R$ edge length at its centre surrounding the rotor. The mesh spacing in this region is connected to the number of

15 blade sections of the AL - $\Delta x = R/(N_s + 1)$. In total 64 ($N_s = 9$) and 128 ($N_s = 19$) cells discretise the flow domain along each dimension, resulting in $2.5 \times 10^5$ and $2.1 \times 10^6$ degrees of freedom. The boundaries off the main flow direction are of the Symmetry type, whereas the inflow and outflow faces obey Dirichlet and Neumann conditions, respectively. The wing test cases follow the same approach only with $N_s = 32$ and an inner box edge length of $3b$, where $b$ is the wing's half-span. This results in 80 cells along each dimension and $5.1 \times 10^5$ degrees of freedom.

The sensitivity of the blade force distribution to the domain size, time step and grid size is explored in Fig. 2. The length of the domain edges is doubled to $50R$, time step and grid size are halved with respect to a setup obeying the method described above. As reference acts a simulation of the NREL 5MW at $14\,\mathrm{ms}^{-1}$ with $N_s = 9$ and $\epsilon = 0.2R$. Though non-zero, the sensitivity of the results is acceptable in code comparison and should impact AL simulations with and without correction similarly.

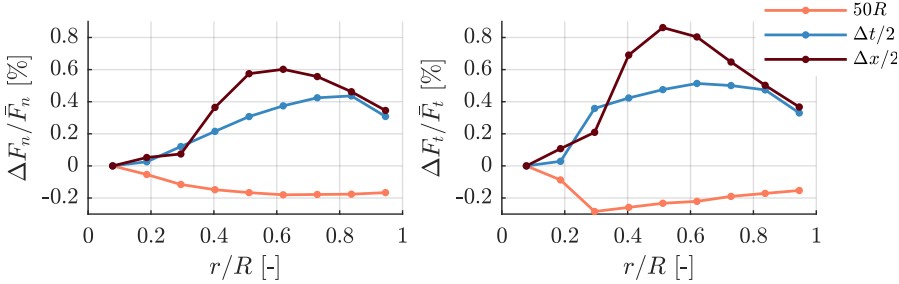

**Figure 2.** Change in normal and tangential forces - normalised by the respective blade average - on the NREL 5MW blades at $14\,\mathrm{ms}^{-1}$ wind speed when doubling the domain size, halving the time step or grid size in the AL simulations. ($N_s = 9$, $\epsilon = 0.2R$, $\bar{F}_n = 2.60 \times 10^3\,\mathrm{Nm}^{-1}$, $\bar{F}_t = 7.38 \times 10^2\,\mathrm{Nm}^{-1}$)



### 3.2 Free-wake lifting-line rotor simulations

The in-house solver MIRAS has been employed to perform the free-wake lifting-line simulations. MIRAS is a multi-fidelity computational vortex model, which is mainly used for predicting the aerodynamic behaviour of wind turbines and its wakes. It has been developed at DTU during the last decade and it is extensively validated for small to large size wind turbine rotors by
Ramos-García et al. (2017, 2014a, b).

The free-wake vortex method essentially models the wake of a wind turbine by a bundle of infinitely thin vortex filaments. To avoid numerical singularities, it is required to introduce a viscous core, which represents a more physical distribution of the velocities induced by each vortex filament, desingularizing the Biot-Savart law near the center of the filament. The velocity induced by each one of the elements is obtained directly by evaluating the Biot-Savart law, and by summing the velocity
induced by all filaments, the total wake induction is obtained.

$$\boldsymbol{u}(\boldsymbol{x}_i) = \sum_{j=1}^{N} K_{ij} \frac{\gamma_j}{4\pi} \frac{\boldsymbol{t}_j \times \boldsymbol{r}_{ij}}{r_{ij}^3} \qquad \text{where} \qquad K_{ij} = \frac{r_{ij}^2}{\left(\varepsilon_j^{2z} + r_{ij}^{2z}\right)^{1/z}} \tag{12}$$

where $N$ is total number of filaments that form the wake, $\boldsymbol{r}_{ij} = \boldsymbol{x}_i - \boldsymbol{y}_j$ is the distance vector from the vortex element $\boldsymbol{y}_j$ to the evaluation point $\boldsymbol{x}_i$. $\gamma_j$ is the circulation of the filament, $\boldsymbol{t}_j$ is the unit orientation vector of the $j$-th filament and $r_{ij} = |\boldsymbol{r}_{ij}|$. $\varepsilon_j$ is the vortex core radius of the filament and $z$ defines the cut-off velocity profile where the Lamb-Oseen model (Lamb, 1932;
Oseen, 1911), $z = 2$, has been employed.

A viscous core model is applied to emulate the effect of viscosity by changing the vortex core radius as a function of time Leishman et al. (2002)

$$\varepsilon_i(t) = \sqrt{4\alpha_v \delta_v \nu t_i} + \varepsilon_0 \tag{13}$$

Here $\alpha_v$ is a constant set to 1.25643 (Ananthan and Leishman, 2004), $\nu$ is the kinematic viscosity and $t_i$ is the time elapsed
since the generation of the $i$-th filament. In order to represent the diffusive time scales, the viscous core radius is set to change with the vortex age by adding a turbulence eddy viscosity, $\delta_v$, first proposed by Squire (1965), and in this work set to $10^{-3}$. To avoid the singular behaviour of newly released vortex elements, an initial core radius, $\varepsilon_0$, is introduced. In accordance with Ramos-García et al. (2017), where it was found out that a small core radius is necessary to have flow convergence, a core radius of 0.1% the local chord at the release station is used.

For the sake of the present study, two different approaches to compute the angle of attack have been followed.

- Inviscid (LL), where the non-regularized Biot-Savart law is used to compute the induction from the wake filaments at the quarter-chord location. This is the standard method used in a lifting line solvers.

- Viscous (LL+core), where the regularized Biot-Savart law is used to compute the induction at the quarter-chord location. A viscous core with radius equal to the actuator line smearing factor is used for a direct comparison of the methods.





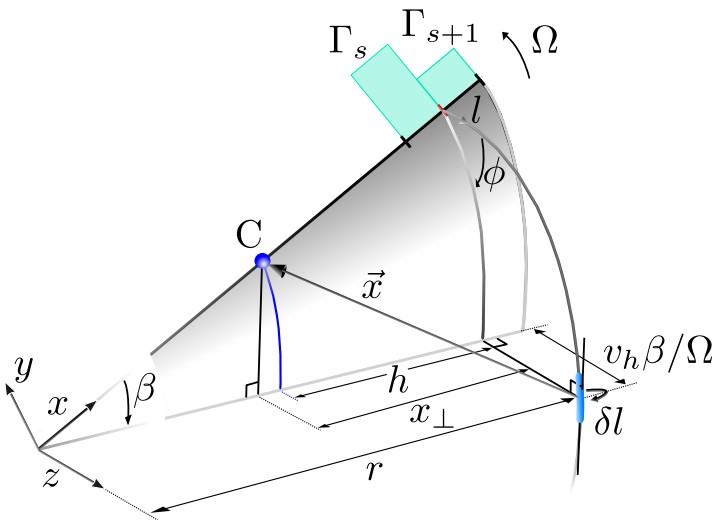

**Figure 3.** Trailed vorticity path. The blade rotates in the $x$-$y$ plane and $z$ points downstream. The vortex element $\delta l$ with strength $\Gamma_s - \Gamma_{s+1}$ is shed at $r$ and transported downstream by the local velocity. The distance from the shedding location $r$ to a point $C$ along the blade is $h$ ($h = r - C_x$), where $\delta l$ induces tangential and axial velocities.

This makes possible a double validation of the models. On one hand the corrected actuator line simulations can be validated against the LL calculations, and on the other hand the raw actuator line model, without tip correction, can be compared against the LL+core simulations which include the smearing effect in the free-wake model.

## 4   Tip/Smearing correction for the actuator line

5   Applying the velocity correction methodology introduced in Sect. 1.1 in three-dimensional space yields a velocity correction vector. The viscous core behaviour of the AL bound vorticity - proven in Sect. 2 to originate from the force smearing - is inherited by the trailing vortices, as will be demonstrated in Sect. 5.1.1. Therefore the induction from the trailed voriticity at the blade is lower than without force smearing. Fig. 3 shows the path of trailed vorticity shed from in-between two sections of a blade with a strength of $\Delta \Gamma = \Gamma_s - \Gamma_{s+1}$ with

10   $$\Gamma_s = \frac{1}{2}\sqrt{v_s^2 + w_s^2}C_l(\alpha)c \quad \phi = -\tan^{-1}\left(w_s/v_s\right) \tag{14}$$

Here $s$ defines the blade section index, $l$ is the vortex following coordinate and $\alpha$ the angle-of-attack, which depends on the vortex inflow angle/helix pitch $\phi$ in combination with blade pitch and twist at the section. The missing induction from this single trailed vortex at a point C is obtained by integrating along the vortex line

$$\boldsymbol{u}^{\star} = \int_{0}^{\infty} f_{\epsilon}\boldsymbol{\delta}\tilde{\boldsymbol{u}}\,\mathrm{d}l \tag{15}$$





Here $\boldsymbol{\delta\tilde{u}}$ is the velocity induced by an infinitesimal element $\boldsymbol{\delta l}$ of the vortex line at point C, which is given by the Biot-Savart law:

$$\boldsymbol{\delta\tilde{u}} = \frac{\Delta\Gamma}{4\pi}\frac{\boldsymbol{\delta l} \times \boldsymbol{x}}{|\boldsymbol{x}|^3} \tag{16}$$

where $\boldsymbol{x}$ is the vector pointing from the element towards C. The smearing factor for this vortex element becomes

$$f_\epsilon = \exp\left(-\frac{(\boldsymbol{x}\hat{\boldsymbol{e}}_\perp)^2}{\epsilon^2}\right) \tag{17}$$

The viscous core only acts in the plane orthogonal to the vortex element $\boldsymbol{\delta l}$, hence $\hat{\boldsymbol{e}}_\perp$ projects $\boldsymbol{x}$ onto this plane. Using the distance here instead $|\boldsymbol{x}|$ as Dag et al. (2017) proposed, violates the two-dimensional nature of the viscous core.

The total missing induction at a blade section $s$ is obtained by summing the contribution from all trailed vortices. The number of trailed vortices $N_v$ is directly related to the number of blade sections $N_v = N_s + 1$. Discretising the vortices in time, the missing induction at a certain blade section becomes

$$\boldsymbol{u}_s^\star = \sum_v^{N_v}\sum_n^{N_t} f_{s,v}^n \boldsymbol{\Delta\tilde{u}}_{s,v}^n \tag{18}$$

Here $v$ denotes the trailed vortex index, $n$ the time index and $N_t$ the number of time steps. Note that $n = 1$ is the most recently shed vortex element. As a tip/smearing correction should remain computationally inexpensive, numerically solving the Biot-Savart law in Eq. 16 to obtain $\boldsymbol{\Delta\tilde{u}}_{s,v}^n$ is unfeasible. This would necessitate $N_t N_v N_s$ or $N_t(N_s+1)N_s$ evaluations. An accurate, yet fast alternative to solving the Biot-Savart law directly is the near-wake model (NWM) for trailed vorticity by Pirrung et al. (2016, 2017b), which also includes downwind convection. It performs well for dynamic flow cases and exhibits great numerical stability as it was originally developed to enhance the aerodynamic accuracy of BEM models. Its formulation is based on a lifting line representation of the blade's trailed vorticity as in Fig. 3 and approximates the induced velocities from a single trailed vortex line by two indicial functions. The velocity induced by a vortex element is given in the NWM as

$$\boldsymbol{\Delta\tilde{u}}_{s,v}^n = \left(X_{s,v}^n + Y_{s,v}^n\right)\begin{bmatrix} 0 \\ \sin(\phi^n) \\ -\cos(\phi^n) \end{bmatrix} \tag{19}$$

with $\phi$ representing the helix angle and the indicial functions taking the form

$$\{X_{s,v}^n, Y_{s,v}^n\} = a_{\{X,Y\}}\frac{r_v}{4\pi h_s|h_s|}\Delta\Gamma_v^n\phi_{s,v}^{\star^n}\left[1 - \exp\left(-b_{\{X,Y\}}\frac{\Delta\beta^{\star^n}}{\phi_{s,v}^{\star^n}}\right)\right]\exp\left(-b_{\{X,Y\}}\sum_i^{n-1}\frac{\Delta\beta^{\star^i}}{\phi_{s,v}^{\star^i}}\right) \tag{20}$$

The definitions of $a_{\{X,Y\}}, b_{\{X,Y\}}, \beta^\star$ and $\phi^\star$ are those of Pirrung et al. (2016, 2017b). The indicial functions allow time-advancing the solution by a mere multiplication, considerably reducing the model evaluations to $N_v N_s + N_v$. In the original formulation this removes the need for bookkeeping, however as the smearing factor also changes with the position of the vortex element all previously shed elements are advanced individually in this specific implementation. This is only an experimental feature for testing the smearing correction and should be simple to remove in a future, practical implementation.




Following the lifting line formulation of the NWM shown in Fig. 3 the perpendicular distance from the vortex element to C becomes

$$x_\perp = \boldsymbol{x}\hat{\boldsymbol{e}}_\perp = \begin{pmatrix} -r\cos(\beta)+r-h \\ r\sin(\beta) \\ v_h\beta/\Omega \end{pmatrix} \begin{pmatrix} -\cos(\beta) \\ \sin(\beta) \\ 0 \end{pmatrix} = r\left[1-(1-h/r)\cos(\beta)\right] \tag{21}$$

Thus the smearing factor becomes

$$f_{s,v}^n = \exp\left(-\frac{r_v^2\left[1-(1-h_s/r_v)\cos(\beta^n)\right]^2}{\epsilon^2}\right) \tag{22}$$

When discretising in time $\beta^n$ is taken to be the mid-point.

Finally the missing velocities computed in Eq. 18 correct the original velocities from the CFD simulations

$$\boldsymbol{u}_s = \boldsymbol{u}_s^{\mathrm{CFD}} + \boldsymbol{u}_s^\star \tag{23}$$

The correction therefore influences the blade forces through the angle-of-attack and the velocity magnitude. It also changes the
10 circulation at each blade section through Eq. 14 and thus the shed vorticity and its induction. Hence determining the correction velocity is an iterative procedure. We use the technique by Pirrung et al. (2017a) established for the NWM to accelerate and ensure its convergence. Furthermore the activation of the correction is delayed until the starting vorticity of the rotor has been transported at least one blade length away from the rotor plane. This enhances its numerical stability, as induction has already build up at the blade by its time of activation.

# 5  Results

## 5.1  Basic wing test cases

To verify the smearing hypothesis presented (Sect. 1.1) and the novel smearing correction (Sect. 4) two basic wing flow cases with known theoretical solutions are modelled. Either a rectangular or elliptic wing is represented by an AL as shown in Fig. 4, where the coordinate system is unchanged from the definition in Fig. 3. The AL is discretised in uniformly spaced sections
in-line with the underlying flow grid and the smearing parameter is twice the section width, which ensures a continuous force distribution along the wing. Is the smearing correction activated, vortices are trailed in-between sections. The common simulation parameters are given in Table 1. Unless specifically stated the sectional lift coefficient $C_l = 1$ and drag is zero along the wing, independent of the angle-of-attack. The chord of the rectangular wing is set to 1m and the elliptical chord distribution is

$$c(x) = c_0\sqrt{1-\left(\frac{2(x-r_{v=1})}{b}\right)^2} \tag{24}$$

with the root chord $c_0 = 4$m. All simulations are performed within the same computational domain, defined in Sect. 3.1.





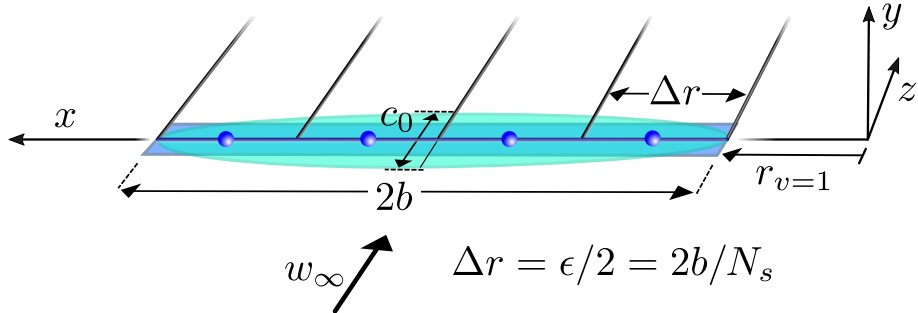

$$\Delta r = \epsilon/2 = 2b/N_s$$

**Figure 4.** Definition of the wing test cases with either a rectangular or elliptic planform. Vortices are trailed in-between sections and the actuator line forces are computed and exerted at the sections' centres.

**Table 1.** Input parameters common to both rectangular and elliptic wing simulations.

| $w_\infty$ [ms$^{-1}$] | $2b$ [m] | $r_{v=1}$ [m] | $\Omega$ [rads$^{-1}$] |
|---|---|---|---|
| 10 | 10 | 0.5 | 0 |

The theoretical predictions of the the velocity field are achieved by representing the vortex system of Fig. 4 with vortex filaments. The velocity induced by a filament with a viscous vortex core at an arbitrary point C is

$$\boldsymbol{u} = f_\epsilon(x_\perp) \frac{\Gamma}{4\pi} \frac{(x_1 + x_2)(\boldsymbol{x_1} \times \boldsymbol{x_2})}{x_1 x_2 + \boldsymbol{x_1} \cdot \boldsymbol{x_2}} \tag{25}$$

$$x_\perp = \frac{|\boldsymbol{x_1} \times \boldsymbol{x_2}|}{|\boldsymbol{x_2} - \boldsymbol{x_1}|} \quad \text{and} \quad x_i = |\boldsymbol{x_i}| \tag{26}$$

where $\boldsymbol{x_1}$ points from the start of the filament to C and $\boldsymbol{x_2}$ from its end. For a definition of $f_\epsilon(x_\perp)$ refer to Eq. 17; without viscous core $f_\epsilon(x_\perp) = 1$. The contribution from different segments is summed to give the overall velocity field. As the wing is lightly loaded we assume all vortex segments to remain in the $x$-$z$ plane.

### 5.1.1   Trailed vorticity smearing

The vortex smearing hypothesis assumes the trailed vorticity inheriting the smeared velocity field from the bound vortex. This
is tested by simulating a rectangular wing without any correction. All vorticity is shed from the wing tips, creating the well known horseshoe vortex. For the hypothesis to be valid the velocity distribution in the plane orthogonal to the trailed vortices should thus be identical to the one of the bound vortex .

Fig. 5 compares the velocities induced by a rectangular wing predicted by an AL and three vortex segments (one bound, two trailed) for five different smearing parameters. Only half of the wing is presented, due to symmetry. Velocity distributions are
shown for lines cutting the vortex segments at right angles for $y = 0$. Clearly the velocity smearing is identical between trailed

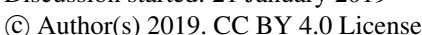



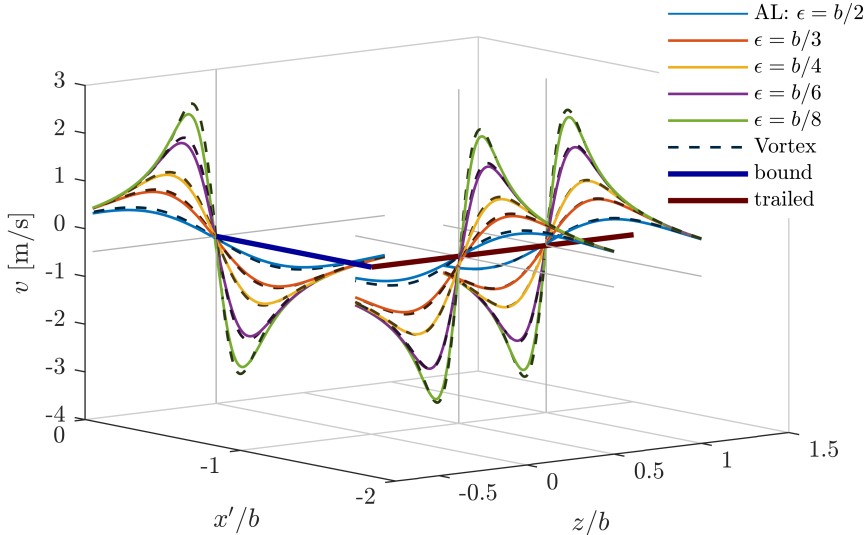

**Figure 5.** Velocities induced perpendicular to a rectangular wing predicted by an actuator line (AL) without correction and by vortex segments with a viscous core (Vortex) with different smearing parameters. Velocities are shown along lines cutting the bound and trailed vorticies at right angles and $y = 0$. Only half of the horseshoe vortex is depicted as $x' = x - (b + r_{v=1})$.

and bound vorticity, confirming the smearing hypothesis. Slight differences are linked to the numerical discretisation of the Gaussian force projection (Shives and Crawford, 2013) and numerical diffusion.

### 5.1.2 Smearing correction verification

As mentioned in Sect. 4, the new smearing correction uses a lifting-line representation of the trailed vorticity. Thus the predic-
tion of the velocity correction with our model or vortex segments should be identical. To simultaneously verify its numerical implementation, our model receives solely the sampled velocities from the flow domain to compute the circulation at the sections. The body forces are not applied inside the domain, though, to avoid influencing the trailed vortex paths nor are the correction velocities added to the CFD velocities to keep the circulation unchanged. This holds the trailed vortices in the $x$-$z$ plane, simplifying the representation of the wake with vortex segments. The segments' circulation is exactly the same as in
the smearing correction to avoid any numerical effects influencing the comparison. Fig. 6 compares the velocity correction predicted by the (ana)lytical vortex segments and our model at each section along a rectangular and elliptic wing for different smearing factors (i.e. $\epsilon = 2b/(N_v-1)$). With decreasing force smearing the velocity correction concentrates towards the tips, as the induced velocity gradients are increasing. Therefore even at higher resolution the smearing correction remains significant, yet more localised. Generally the model slightly over-predicts the missing induction at the wing, becoming more prominent
with increasing resolution. With $N_v = 64$ the difference maximally reaches $6.7\%$ (rectangular) and $1.7\%$ (elliptic) with respect to the inflow velocity. The average error does not breach $0.5\%$ in any case. The velocity jump towards the tip sections of the elliptical wing is related to the equidistant discretisation of the wing (Pirrung et al., 2014).

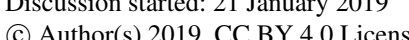



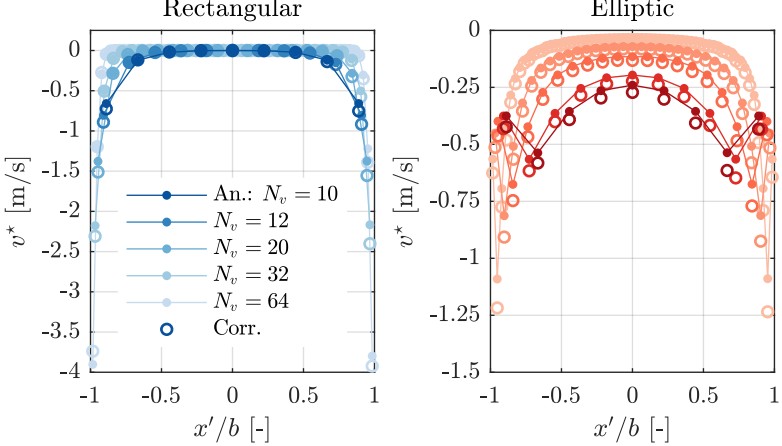

**Figure 6.** Analytical (An.) and corresponding model prediction of the velocity correction for varying force smearing ($\epsilon = 2b/(N_v - 1)$) along a rectangular and elliptic wing, where $x' = x - (b + r_{v=1})$.

### 5.1.3 Coupled AL-smearing correction verification

The coupling between velocity correction and the flow domain is verified by comparing the corrected downwash at an elliptical wing to the theoretical expectation. The downwash should be constant along the wing and is given by

$$v_{th} = -\frac{\Gamma_0}{4b} = -\frac{w_\infty c_0 C_l}{8b} \tag{27}$$

where $\Gamma_0$ is the circulation at the wing root. Similar to Shives and Crawford (2013) the $C_l$ was not fixed, but instead followed the theoretical lift curve slope for thin airfoils $C_l = 2\pi$. For the wing to operate at a constant lift coefficient $C_l = 1$, its angle-of-attack needed to include the effect of the induced velocities:

$$\alpha = \alpha_{eff} + \alpha_i = \frac{C_l}{2\pi} + \tan^{-1}\left(\frac{c_0 C_l}{8b}\right) \tag{28}$$

This represents are more rigorous test of the coupled system then prescribing the loading along the wing, as only the correct
downwash leads to the desired, constant sectional lift coefficients.

Fig. 7 shows the downwash predicted by AL simulations with different smearing parameters and active correction. The CFD component of the velocities are shown ($v^{CFD}$) separately to emphasise the contribution of the correction to arrive at the correct, constant downwash of $1 \text{ ms}^{-1}$. Clearly without the correction, the induced velocities are a function of the smearing factor and only arrive at the theoretically expected value for $N_v = 32$. Including the correction greatly reduces the dependence of the
downwash on the force smearing. The insufficient correction towards the tips feeds back to the equidistant discretisation of the AL (Pirrung et al., 2014), which is linked to the uniform spacing of the underlying flow grid.





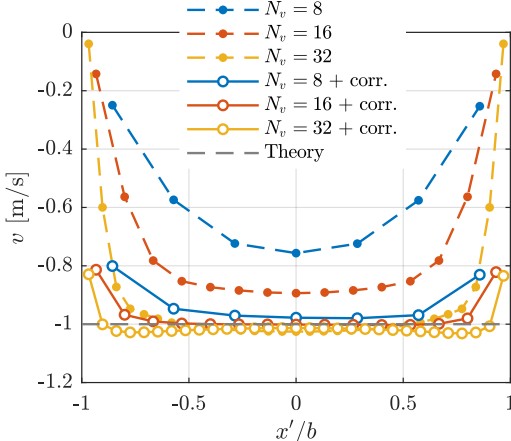

**Figure 7.** Downwash at an elliptical wing predicted by AL simulations with different smearing factors and smearing correction. The CFD component of the velocities are shown ($v^{\mathrm{CFD}}$) as well as the total downwash incorporating the correction ($v^{\mathrm{CFD}} + v^{\star}$). The theoretical value acts as reference.

## 5.2   Rotor simulations - NREL 5MW

The validity of the smearing hypothesis and its correction in rotor applications is demonstrated with simulations of the NREL 5MW turbine (Jonkman et al., 2009) using actuator (AL) and lifting line (LL) models. The input parameters to these simulations are given in Table 2.

**Table 2.** Input parameters to the NREL 5MW simulations.

| $V_\infty$ [ms$^{-1}$] | $\Omega$ [rpm] | Pitch [°] |
|---|---|---|
| 4 | 4.6 | 0.00 |
| 6 | 6.9 | 0.00 |
| 8 | 9.2 | 0.00 |
| 14 | 12.1 | 2.59 |
| 25 | 12.1 | 23.09 |

5   ### 5.2.1   Uniform inflow

Fig. 8 compares the AL results with and without the novel smearing correction to the LL with and without viscous core. At this wind speed of 8 ms$^{-1}$ the thrust coefficient is highest ($C_T = 0.84$) - and hence induction - thus lending itself as a strong verification case. Clearly, there is an equivalence between the original AL and the LL with a viscous core and the corrected AL with the LL. The smearing correction thereby makes the AL effectively act as a LL, as originally intended by Sørensen




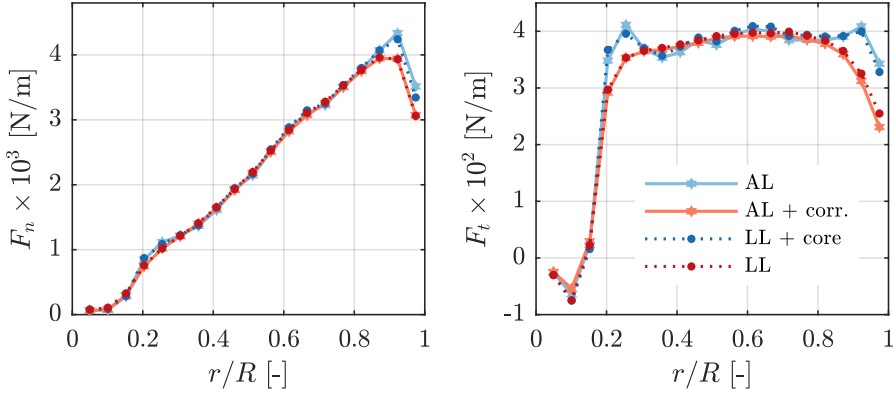

**Figure 8.** Normal and tangential forces on the NREL 5MW blades at 8 ms$^{-1}$ predicted by AL simulations with/without smearing correction and LL with/without viscous core. ($N_s = 19$, $\epsilon = 0.1R$)

and Shen (2002). The impact of the viscous core is most prominent towards blade root and tip. The sudden drop in the forces predicted by the AL/LL+core for the tip section of the blade - located at $r/R = 0.97$ - is not triggered by any aerodynamic tip effects, but relates to a pronounced reduction in chord. While not greatly affecting the magnitude of the forces in the mid-section of the blade, the viscous core does introduce greater fluctuations in the force distribution. The missing induction
introduced by the viscous core hence reduces the coupling between neighbouring blade sections. The smearing correction also recovers this behaviour of the LL.

Surpassing rated wind speed, forces increase inboard until cut-out. Just before cut-out at 25 ms$^{-1}$ loading thus reaches a maximum towards the root, causing an equally pronounced influence of the smearing correction in this region as demonstrated in Fig. 9. Again, the equivalence of the AL and LL implementations is remarkable. This high wind-speed case also demonstrates
our correction is not only a *tip correction*.

The comparison of AL and LL is summarised in Fig. 10 in the form of local thrust and power distributions at different wind speeds. Note that for the wind speeds below rated ($< 11.4$ ms$^{-1}$), the coefficients are identical. The results are only presented for simulations with $N_s = 9$ for visibility, but compare equally well at higher resolution. As mentioned earlier, the smearing corrections acts predominantly towards tip and root. An additional overview of all results is given in Table 3. Here the total
rotor thrust $T$ and power $P$ predicted by the corrected actuator line (AL$^\star$) and the lifting line (LL) are listed as well as the influence of adding the viscous core relative to AL$^\star$ and LL, respectively. The AL and LL solutions are not directly compared to avoid including any mean bias in the comparison. The impact of the correction on the AL forces is nearly identical to removing the viscous core in the LL simulations at any wind speed, which further supports our correction methodology.





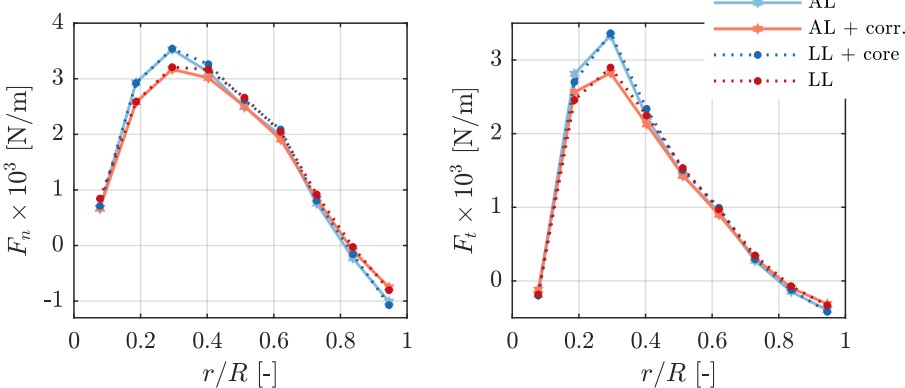

**Figure 9.** Normal and tangential forces on the NREL 5MW blades at 25 ms$^{-1}$ predicted by AL simulations with/without smearing correction and LL with/without viscous core. ($N_s = 9$, $\epsilon = 0.2R$)

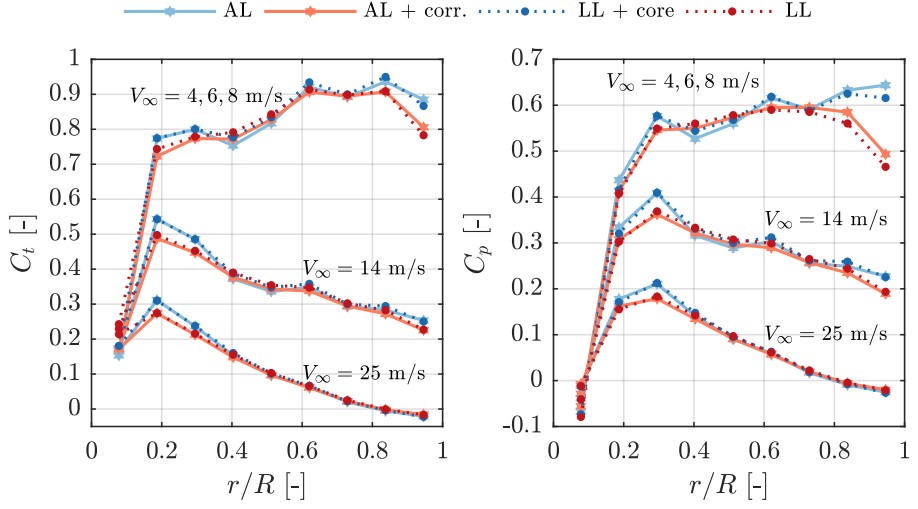

**Figure 10.** Local thrust and power coefficients along the NREL 5MW blades at different wind speeds predicted by AL simulations with/without smearing correction and LL with/without core. ($N_s = 9$, $\epsilon = 0.2R$)

#### 5.2.2 Yawed inflow

As the smearing correction does not include yaw effects - the wake is assumed to advect normal to the rotor plane - we tested its influence at yaw angles $\chi$ of $15, 30$ and $45$ degrees at $8$ ms$^{-1}$. Again the LL with and without viscous core acted as reference. The time steps remained the same as in uniform inflow. Here only the results for the most extreme case at $45°$ yaw are shown, as then the differences are most severe. Fig. 11 presents the normal and tangential force variation during one rotation, averaged over three distinct regions of the blade, at a wind speed of $8$ ms$^{-1}$. Whilst the agreement is best towards the blade tip, the


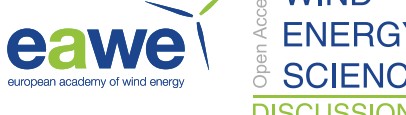

**Table 3.** An overview of the simulation inputs and results for the NREL 5MW in uniform inflow. Results are grouped by blade/grid resolution. For the actuator line (AL) and lifting line (LL) the simulation time step $\Delta t$, the total thrust $T$ and power $P$ and the relative change in these quantities caused by the correction/removing the viscous core are listed. Note that AL$^\star$ represents the corrected AL results and change is expressed relative to the rotor thrust and power.

| $N_s$ | $\epsilon/R$ | $V_\infty$ [ms$^{-1}$] | $\Delta t \times 10^{-2}$ [s] | | $T \times 10^5$ [N] | | $\Delta T$ [%] | | $P \times 10^6$ [W] | | $\Delta P$ [%] | |
|---|---|---|---|---|---|---|---|---|---|---|---|---|
| | | | AL | LL | AL$^\star$ | LL | $\Delta$ AL | $\Delta$ LL | AL$^\star$ | LL | $\Delta$ AL | $\Delta$ LL |
| 9 | 0.2 | 4 | 15.90 | 18.12 | 1.01 | 1.02 | 4.26 | 4.02 | 0.26 | 0.26 | 10.6 | 9.67 |
| | | 6 | 13.82 | 12.08 | 2.27 | 2.30 | 4.30 | 4.02 | 0.89 | 0.87 | 10.7 | 9.66 |
| | | 8 | 10.36 | 9.06 | 4.08 | 3.93 | 3.67 | 4.02 | 2.13 | 2.05 | 9.1 | 9.66 |
| | | 14 | 7.87 | 6.89 | 4.64 | 4.77 | 4.96 | 4.51 | 5.39 | 5.54 | 7.03 | 6.04 |
| | | 25 | 7.87 | 6.89 | 2.84 | 2.99 | 10.1 | 10.2 | 5.40 | 5.68 | 12.7 | 11.9 |
| 19 | 0.1 | 4 | 10.37 | 18.12 | 0.98 | 1.02 | 3.56 | 3.66 | 0.25 | 0.25 | 8.89 | 8.87 |
| | | 6 | 6.90 | 12.08 | 2.21 | 2.30 | 3.55 | 3.64 | 0.83 | 0.86 | 8.87 | 8.83 |
| | | 8 | 5.17 | 9.06 | 3.92 | 3.95 | 3.56 | 2.93 | 1.98 | 2.02 | 8.88 | 6.85 |
| | | 14 | 3.93 | 6.89 | 4.52 | 4.76 | 4.11 | 3.97 | 5.22 | 5.51 | 6.09 | 5.72 |

variation in the force with blade position is similar between AL and LL simulations for any section along the blade. AL results are shifted downwards with respect to the LL predictions at the inner sections, hinting at the AL experiencing higher induction in this region. But for the verification of the smearing correction this shift is irrelevant, instead its impact on the AL forces needs to be judged relative to the difference between LL with and without core. In this respect the smearing correction is
5 behaving correctly, increasing forces in a similar fashion as a LL without core in the mid-section of the blade and reducing them towards the root and tip.

### 5.2.3 Pitch step

The pitch step is defined as

$$\psi = \psi_0 + \frac{\Delta\psi}{2}\left[1 + \tanh(k(t-t_0))\right] \tag{29}$$

with $\psi_0$ defining the pitch angle before the step, $t_0$ the time instant of the step and $\Delta\psi$ the pitch change. Here an extremely violent step is chosen - determined by $k$ - to encourage an equally pronounced blade force response. The parameters governing this comparison are given in Table 4, which realise a pitch step of $\pm 2°$ in $0.14\,\mathrm{s}$ (10% to 90% pitch). To capture the swift change in pitch, the time step is adjusted in both AL and LL simulations to $3.94 \times 10^{-2}\,\mathrm{s}$. The blade force response is normalised as

$$\hat{F}(t) = \frac{F(t) - F_0}{F_\infty - F_0} \tag{30}$$

with $F_0$ and $F_\infty$ denoting the steady-state values before and after the pitch step, respectively.

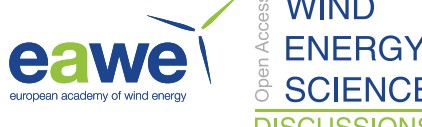



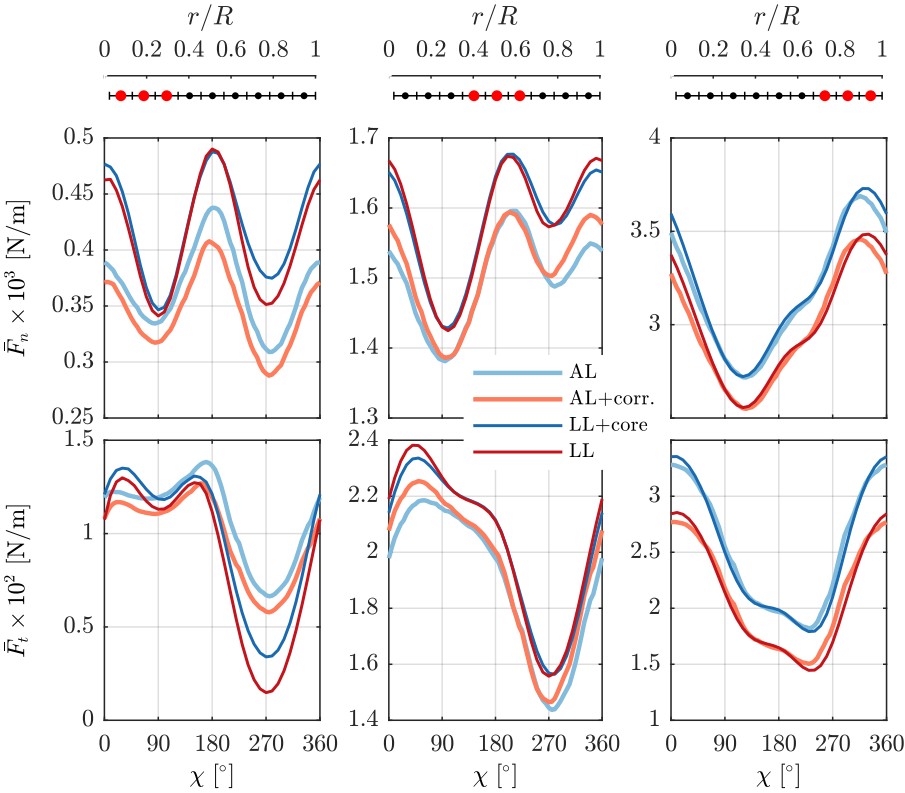

**Figure 11.** Normal (upper panels) and tangential (lower panels) force variation during one rotation of the NREL 5MW blades at $8\,\mathrm{ms}^{-1}$ and $45°$ yaw. The forces are averaged over three sections of the blade and are predicted by AL simulations with/without smearing correction and LL with/without viscous core. The blade is facing upwind at $\chi = 0°$ and is pointing vertically up at $\chi = 90°$. ($N_s = 9$, $\epsilon = 0.2R$)

Here only the tangential force response after a $+2°$ step for the mid and tip blade sections are shown in Fig. 12, as they capture the main features of the response. As the definition here is positive pitch to feather, the force decreases along the blade for positive pitch changes. The AL simulations exhibit a faster response with a kink at 0.14 s, coinciding with the pitch change reaching 99% of the step. The AL therefore seems to capture the pitch rate lift. The LL does not show this feature so - as in yaw - the correct behaviour of the smearing correction on the AL force response should be assessed relative to the influence of removing the viscous core in the LL model. Overall, the correction has limited effect on the dynamic response, which is also confirmed by the LL simulations, but the correction essentially acts on the forces as removing the core in the LL. In the mid-section it reduces the forces by maximally 1% during the first 2 s, dropping to 0.5% afterwards. At the tip section it intensifies the response by maximally 1%, diminishing to 0.3% at 4 s.





**Table 4.** Inputs defining the pitch step.

| $V_\infty$ [ms$^{-1}$] | $\Omega$ [rpm] | $\psi_0$ [°] | $\Delta\psi$ [°] | $k$ |
|---|---|---|---|---|
| 14 | 12.1 | 2.59 | $\pm 2$ | 16 |

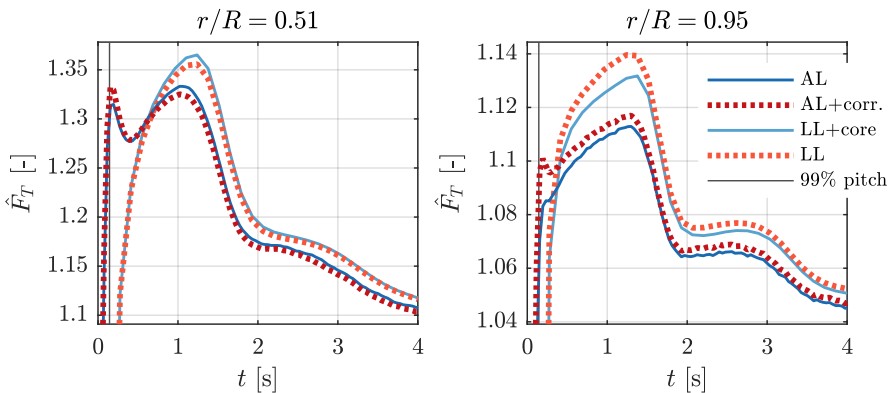

**Figure 12.** Normalised tangential force response at two blade sections (middle and tip) of the NREL 5MW following a pitch step of $+2°$ in $0.14\,\mathrm{s}$ at $14\,\mathrm{ms}^{-1}$ predicted by AL simulations with/without smearing correction and LL with/without viscous core. ($N_s = 9$, $\epsilon = 0.2R$)

### 5.2.4 Turbulent inflow

Highly turbulent inflow should challenge the numerical stability of the new smearing correction by introducing strong and abrupt changes in the angle-of-attack. Comparing simulations with and without inflow turbulence should furthermore reveal, whether turbulence alters the nature of the correction. Fig. 13 shows the impact of the smearing correction on the time-averaged normal and tangential blade forces at $8\,\mathrm{ms}^{-1}$ mean wind speed for AL simulations with uniform and turbulent inflow. With a turbulence intensity (TI) of $15\%$, the forces are unsurprisingly slightly larger ($\approx 20\,\mathrm{Nm}^{-1}$) than in uniform inflow. However the change in forces introduced by the correction is nearly identical ($< 2\,\mathrm{Nm}^{-1}$). When comparing the standard deviation of the forces with and without correction in Fig. 14 the smoothing and dampening effect of the smearing correction on the forces also in highly turbulent inflow becomes apparent. Madsen et al. (2018) observed a corresponding reduction of the force variations on the whole rotor blade when comparing near wake model results against BEM results for the NM 80 rotor in turbulent inflow. This illustrates that the smearing correction leads to the same dynamic coupling between neighbouring blade sections as a lifting line model.

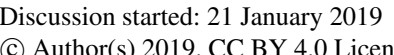



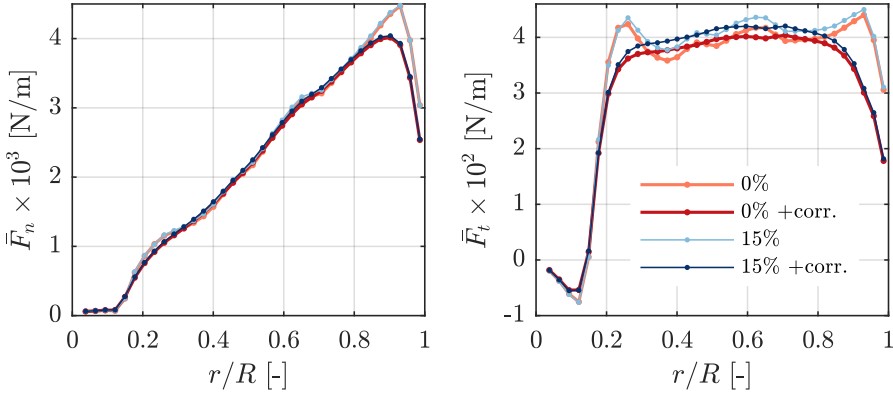

**Figure 13.** Time-averaged normal and tangential forces on the NREL 5MW blades at 8 ms$^{-1}$ mean wind speed and changing inflow turbulence predicted by the AL model with and without smearing correction. ($N_s = 35$, $\epsilon = R/8$)

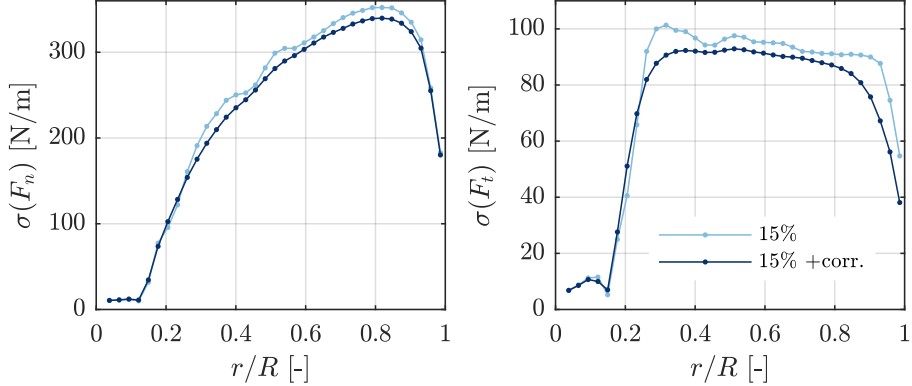

**Figure 14.** Variation of normal and tangential forces on the NREL 5MW blades at 8 ms$^{-1}$ mean wind speed and turbulence intensity of 15% predicted by the AL model with and without smearing correction. ($N_s = 35$, $\epsilon = R/8$)

## 6   Conclusions

The actuator line was intended as a lifting line technique for CFD applications. In this paper we proof - theoretically and practically - that smearing the forces of the actuator line in the flow domain necessarily leads to smeared velocity fields. For the typical Gaussian force projection the widely known Lamb-Oseen (Lamb, 1932; Oseen, 1911) viscous core appears in both bound and trailed vorticity. This core reduces the velocities approaching the vortex centre compared to the inviscid solution of the lifting line. The trailed vorticity of an actuator line thus induces lower velocities at the blade owing to the force projection. We recover this missing induction by combining a near-wake model of the trailed vorticity with Lamb-Oseen's viscous core model and coupling it with the actuator line model. Basic wing test cases with theoretical solutions verify the correction, as





it recovers nearly all induction independent from the severity of the force smearing. Rotor simulations furthermore show the applicability and strength of the correction over the entire operational wind speed range as well as in yaw, strong turbulence and undergoing pitch steps. Here the correction is validated with lifting line simulations with and without viscous core, that are representative of an actuator line with and without smearing correction, respectively. The agreement between the respective
actuator and lifting line results is remarkable.

The current implementation of the smearing correction relies on heavy bookkeeping. In future versions the latter will be removed without jeopardising stability nor accuracy, making it suitable for wind farm simulations in realistic atmospheric flows. Potentially, the correction might also enable accurate rotor simulations at lower discretisation.

*Code and data availability.* All data and parts of the code covering the smearing correction are available on request. Commercial and re-
search licenses for EllipSys3D can be purchased from DTU.

*Author contributions.* AMF was responsible for writing the paper, except for Sect. 3.2 which was authored by NRG. The original numerical implementation of the near-wake model by GP was adapted by AMF to incorporate the smearing correction with the help of GP. NRG performed all simulations with the lifting-line and AMF those with the actuator line. All authors decided on the flow cases to simulate and commented on the document.

*Competing interests.* The authors declare no conflict of interest.

*Acknowledgements.* We would like to acknowledge DTU Wind Energy's internal project "Virtual Atmosphere" for partially funding this research. Furthermore many thanks to Senior Researcher Dr. Mac Gaunaa for his insights on vortex aerodynamics and Senior Researcher Dr. Niels Troldborg for his input on actuator line simulations/modelling, both from DTU Wind Energy. Thanks also to Ang Li for his help on the near-wake model.





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
