# Peer review of "A vortex-based tip/smearing correction for the actuator line"

_Wind Energy Science, 2018_

## Referee Comment (RC1) · Anonymous Referee #1 · 22 Feb 2019

General Comments:

The paper presents an correction tot he Actuator Line (AL) model commonly employed for finite volume simulations of wake turbine aerodynamics. An idealized proof is provided showing that a smeared representation of the volume forces of the AL necessarily lead to a smearing of the velocity field, analogous to the effect of making use of a viscous core model in a lifting line (LL) code. The correction is clearly important, and leads to a necessary correction of the AL model, such that the forcing induced by the blade is more realistic, as realistic as a standard LL in fact.

The paper is structured quite well, and most results are easy to follow, some discrepancies in the results are seen and the proof is quite idealized, so the general application to a fully 3D, viscous flow certainly cannot be claimed. The model is presented clearly

and allows the reader to follow the arguments made generally quite well.

Specific Comments: Page 1, Line 2: Would there really be any doubt that a smeared force distribution gives rise to a smeared velocity field? Page 2, Line 11: Why should the AL produce the same velocities as LL? The AL simulation will be generally viscous, as opposed to a lifting line, which (even with core model) really is based upon an inviscid external flow. Page 2, Line 19: Rephrase: "to exhibit the same viscous core behavior"... Page 3, Line 1: Perhaps more logical to use "tangential" velocity instead of vertical. Page 4, Line 1: The assumption that the spanwise flow is negligible is violated at the tip and the blade root, where one expect the largest gradients, hence this proof is only idealized really. Page 4, Line 2: Simply because the BL is not resolved does not necessarily imply that we can ignore viscous effects, especially not for large inflow shear gradients. Page 4, Line 3: Can the assumption of quasi-steadiness be demonstrated with some sort of simple example, perhaps an order of magnitude argument using the Strouhal number perhaps? Simply this statement seems open to attack, especially for situations with large gradients/ flow deflection. Page 4, Line 16: Here tangential velocity is referred to, as opposed to the previous page with "vertical velocity" Page 5, Line 9: Which grid size is referred to here? Are uniform grids generally used for this type of situation. Statement is ambiguous. Page 5, Line 17: Dirichlet and Neumann implies velocity and pressure? Or is there a reason to use the more general terms? Page 6, Line 29: Is there a reason to use a core size equal to the local chord? Otherwise this could appear to be a "tuning" parameter. Is this expected to produce the comparable velocities? Page 9, Line 21: It is stated that upon activation of the smearing correction, vortices are trailed in-between sections. Is this as the model is treated as being steady/converged, and hence should not have trailing vorticity? Page 9, Line 25: Equation appears to be erroneous: Suggest removing (2) from bracketed term in radical Page 15, Figure 10: Could the author please provide slightly more explanation as to why the differences at the root and tip are so drastic? Is the legend of Figure 9 the same as Figure 9? Page 16, Table 3: This results appears to be drastic as opposed to the AL. Shall the reader then assume that AL results using Ellypsis 3D thus far have
been out by such a factor? Page 16, Line 1: "for any section along the blade." Is the correct color scheme used for these plots, as it appears that the trend is reversed in the tip region. Page 16, Line 3: This doesn't appear to be true for the downwind section of the yawed turbine. Page 17, Line 4: Is this statement made as the "return" to F_inf results appears to set in quicker? Page 17, Line 6: If the physical effect of smearing is inspected, does one necessarily expect a large impact on dynamic response? Page 19, Figure 13: Check color scale again. Results appear to not coincide with text (or I have interpreted them wrong). Page 19, Line 5: Restatement of reviewers earlier statement regarding inviscid approach of LL. Page 20, Line 1: indendepent of

Technical Corrections: Page 1, Line 1: "prove," not proof (repeated on Page 19, Line 2:) Page 1, Line 5: covers Page 1, Line 8: lifting Page 1, Line 21: over-predicted, meaning Page 5, Line 17: symmetry Page 7, Line 7: vorticity Page 8, Line 7: instead of |x| ... Page 10, Line 9: inherits Page 11, Figure 5: Suggest plot color change/line style change. Bund trailed not differentiable.In general plot is also not necessarily well displayed like this. One plot for streamwise and one for spanwise would work better (reviewers opinion!) Page 11, Line 11: (ana)lytical? Page 12, Figure 6: Expand plot width and choose different line style/color style. Results can barely be differentiated. Page 12, Line 9: a /= are Page 12, Line 9: than

---

## Referee Comment (RC2) · Anonymous Referee #2 · 17 Mar 2019

The paper deals with a correction for the actuator line method. This subject is very interesting, since many codes are using actuator line method for multiscale modelling, in particular in the wind energy field. The paper is written not in a straight forward organization, that does not allows to easily follow the discussion. Some specific comments: - page 5 line 12: there is no figure indicating the domain for the rotor simulation nor the wind test case and there is no indication of the position of the rotor/wing in the domain. - page 5 line 15: maybe I'm missing it, but $N\_s$ is not specified. - page 5 line 22: the sensitivity study for AL should be reported as well since it is interesting as well. - table 3: do you compare with state of the art calculation of NREL5MW actuator line computations?

---

## Author Comment (AC1) · 11 Apr 2019

April 11, 2019

alrf

Dear Reviewers

First of all we would like to thank you for the constructive comments on our article. They were very helpful in improving our paper and hope that you will accept the revised manuscript for publication. Please find below our responses (in black) to your comments (in blue). We added more detail to the numerical methodology and the description of the correction to improve the reproduce-ability of our work. For consistency, we now also use the same grid for all AL simulations independent of the smearing length scale.

Yours sincerely

AR. Meyer Forsting, G. Pirrung and N. Ramos-García

VAT no. DK 30 06 09 46

Technical University of Denmark          Frederiksborgvej 399          Tel   +45 45 25 11 99          alrf@dtu.dk
**Department of Wind Energy**              Building 125                  Dir   +45 93 51 11 75          www.dtu.dk
                                          DK-4000 Roskilde

[Figure]

**Comments to Reviewer #1**
**General Comments**

The paper presents an correction to the Actuator Line (AL) model commonly employed for finite volume simulations of wake turbine aerodynamics. An idealized proof is provided showing that a smeared representation of the volume forces of the AL necessarily lead to a smearing of the velocity field, analogous to the effect of making use of a viscous core model in a lifting line (LL) code. The correction is clearly important, and leads to a necessary correction of the AL model, such that the forcing induced by the blade is more realistic, as realistic as a standard LL in fact. The paper is structured quite well, and most results are easy to follow, some discrepancies in the results are seen and the proof is quite idealized, so the general application to a fully 3D, viscous flow certainly cannot be claimed. The model is presented clearly and allows the reader to follow the arguments made generally quite well.

**Specific Comments**

Page 1, Line 2: Would there really be any doubt that a smeared force distribution gives rise to a smeared velocity field? It seems quite natural, however until now the issue was the exact shape of the smeared velocity profile and how it correlates with the forcing smearing function. We are now more specific: *"In this paper we proof - theoretically and practically - that smearing the forces of the actuator line in the flow domain forms a viscous core in the bound and shed vorticity of the line."*

Page 2, Line 11: Why should the AL produce the same velocities as LL? The AL simulation will be generally viscous, as opposed to a lifting line, which (even with core model) really is based upon an inviscid external flow. In the very limit the AL should approach the LL. By reducing the smearing (and in-line the grid spacing) the force smearing acts eventually as a Dirac delta function and thus should be equivalent to a LL. Of course this is a purely theoretical exercise as a singular force is neither a correct physical representation nor desirable in a CFD context. This of course ignores the effect of viscosity, however in AL simulations the physical vortex core size is usually not attained. Sentence modified to: *"Ignoring viscous effects, the AL should in principle induce the same velocities as a LL - equivalent to the inviscid solution."*

Page 2, Line 19: Rephrase: "to exhibit the same viscous core behavior". . . Changed accordingly.

Page 3, Line 1: Perhaps more logical to use "tangential" velocity instead of vertical. Changed accordingly.

Page 4, Line 1: The assumption that the spanwise flow is negligible is violated at the tip and the blade root, where one expect the largest gradients, hence this proof is only idealized really. True and any LL representation suffers from this. Yet LL methods consistently yield accurate blade forces for large wind turbines (Qblade, MIRAS etc). We changed the sentence and specified the validity of our assumption: *"Away from root and tip, the flow around a high aspect ratio blade is nearly two-dimensional, as the span-wise flow is negligible."*

Page 4, Line 2: Simply because the BL is not resolved does not necessarily imply that we can ignore viscous effects, especially not for large inflow shear gradients. An unfortunate sentence, we

changed it to: *"Viscous effects are disregarded in light of the large Reynolds numbers encountered."*. In the proof we are not claiming that our assumptions are holding in any kind of flow situation. The last sentence of this sections therefore reads: *"This marks the theoretical confirmation of the observations by [], which additionally indicates that a viscous core behaviour with an AL requires inviscid, two-dimensional and locally steady flow conditions."*

Page 4, Line 3: Can the assumption of quasi-steadiness be demonstrated with some sort of simple example, perhaps an order of magnitude argument using the Strouhal number perhaps? Simply this statement seems open to attack, especially for situations with large gradients/ flow deflection. The assumption of quasi-steady flow holds as long as the flow is attached to the airfoil surface and therefore can be modelled with a singular vortex line/sheet leaving the airfoil/blade. As soon as the flow starts to separate the quasi-steady assumption does not hold any longer and the flow dynamics cannot be captured by LL/AL methods. The Strouhal number becomes only relevant once the condition is violated. The text was changed to: *"Furthermore the relationship between body force and flow-field becomes quasi-steady assuming the flow is attached."*

Page 4, Line 16: Here tangential velocity is referred to, as opposed to the previous page with "vertical velocity" Changed to tangential everywhere.

Page 5, Line 9: Which grid size is referred to here? Are uniform grids generally used for this type of situation. Statement is ambiguous. Changed to: *"The smearing length scale is twice the uniform grid size surrounding the rotor as recommended by [] to guarantee numerical stability."*

Page 5, Line 17: Dirichlet and Neumann implies velocity and pressure? Or is there a reason to use the more general terms? We added more detail: *"All variables, except pressure and its correction which necessitate special treatment [], obey Symmetry conditions on the lateral boundaries, whereas at the inflow and outflow faces they follow Dirichlet and Neumann conditions, respectively"*.

Page 6, Line 29: Is there a reason to use a core size equal to the local chord? Otherwise this could appear to be a "tuning" parameter. Is this expected to produce the comparable velocities? The text reads: *"In accordance with Ramos-García et al. (2017), where it was found that a small core radius is necessary to have flow convergence, a **core radius of 0.1% the local chord** at the release station is used."* Usually the core radius employed in free wake vortex methods is of around 10% of the local chord, however in the referred paper we found that such large initial radius affected the model convergence for small azimuthal wake discretizations. With the core radius of 0.1% we got a very good convergence in terms of both blade forces and flow velocities all the way to an azimuthal discretization of the wake of 0.625 degrees (so about 576 time steps per revolution).

Page 9, Line 21: It is stated that upon activation of the smearing correction, vortices are trailed in-between sections. Is this as the model is treated as being steady/converged, and hence should not have trailing vorticity? Trailing vorticity between sections is due to the radial bound vorticity gradient and will thus also appear in steady state. However, we deleted this sentence, as it could be confusing and the correction and its activation was already explained in the previous section. The correction does start with little delay, only to allow the build up of induction at the blade after

inserting the forces in the flow field. At the end of the previous section it reads: *"Furthermore the activation of the correction is delayed until the starting vorticity of the rotor has been transported at least one blade length away from the rotor plane."*

Page 9, Line 25: Equation appears to be erroneous: Suggest removing (2) from bracketed term in radical  We corrected the equation.

Page 15, Figure 10: Could the author please provide slightly more explanation as to why the differences at the root and tip are so drastic? Is the legend of Figure 9 the same as Figure 9? We included a discussion and a figure showing the effect the correction on the angle of attack, which causes the drop towards the root and tip. The legend is the same.

Page 16, Table 3: This results appears to be drastic as opposed to the AL. Shall the reader then assume that AL results using Ellypsis 3D thus far have been out by such a factor?  Firstly this is not related to EllipSys3D specifically, but instead to the AL method in general. All AL implementations, no matter which CFD solver, show this kind of behaviour. Using highly resolved grids reduces the error, but does not eradicate it (see Jha et al. (2014) for instance). However, in practice the AL method is always used in connection with a tip correction (some kind of modified Glauert/Prandtl implementation), which is not valid for AL methods, as we argue in the introduction. The table is used to demonstrate the equivalence of the force smearing and viscous core radius for AL and LL.

Page 16, Line 1: "for any section along the blade." Is the correct color scheme used for these plots, as it appears that the trend is reversed in the tip region.  The color scheme is the same as in the previous plots. Light blue should be compared to dark blue and light red with dark red. We modified the sentence to highlight that we observe the same variation with azimuthal position:*"Whilst the agreement is best towards the blade tip, the force variation with azimuthal position is similar between AL and LL simulations across all sections."* The next sentence clearly mentions that the overall level is not the same, however.

Page 16, Line 3: This doesn't appear to be true for the downwind section of the yawed turbine. There seems to be a misunderstanding of the figure. We averaged forces over three parts of the blade and then show their variation with azimuthal position. We made this clearer in the text and the caption. The normal forces are generally lower over the inner sections, which points towards higher induction in the AL results. The reason is unclear, but irrelevant in the context of our study.

Page 17, Line 4: Is this statement made as the "return" to $F_{\text{inf}}$ results appears to set in quicker? The statement is related to the previous sentence that identifies a kink in the response, when 99% of the pitch step has been executed. There is a lift term proportional to the pitch rate as indicated by 2D unsteady airfoil theory, so while pitching there is increased lift, which stops with the end of the step.

Page 17, Line 6: If the physical effect of smearing is inspected, does one necessarily expect a large impact on dynamic response?  Not at all, however for completeness and to test the numerical stability of our correction we included these pitch step cases as well. We now mention this in the text.

[Figure]

Page 19, Figure 13: Check color scale again. Results appear to not coincide with text (or I have interpreted them wrong).  We adapted the colours to match all previous plots. In these plots its important to focus on the change from AL to AL with correction and LL with core to LL wihtout core. It is only the relatively impact of the correction/core we are focusing on in this case.

Page 19, Line 5: Restatement of reviewers earlier statement regarding inviscid approach of LL. As mentioned earlier in the very limit the AL should approach the LL.

Page 20, Line 1: indendepent of  Changed.

**All other Editorial Comments have been reviewed and incorporated as much as possible.**

[Figure]

**Comments to Reviewer #2**
**General Comments**

The paper deals with a correction for the actuator line method. This subject is very interesting, since many codes are using actuator line method for multiscale modelling, in particular in the wind energy field. The paper is written not in a straight forward organization, that does not allows to easily follow the discussion. We believe - supported also by the comments of the other reviewer - that the organisation of our paper is appropriate for the problem at hand. We are surprised to find no specific comments on this matter that detail in which way the organisation is lacking clarity. However, we have added more detail in the numerical methodology and the presentation of our correction, which could perhaps help follow our arguments.

**Specific Comments**

page 5 line 12: there is no figure indicating the domain for the rotor simulation nor the wind test case and there is no indication of the position of the rotor/wing in the domain. We chose not to include a figure of the computational domain, due to simplicity of the numerical setup. Instead we described in words the numerical domain from lines 12 to 19 on page 5 and also refer to other papers in which we used the same setup. We believe that a figure showing the grid convergence is of greater value to the reader, nevertheless we have added further detail to the numerical method.

page 5 line 15: maybe I'm missing it, but $N_s$ is not specified. We now specify it.

page 5 line 22: the sensitivity study for AL should be reported as well since it is interesting as well. The aim of the paper is not to find the optimal discretisation of the AL, but to show the equivalence between the AL and LL, when the same discretisation is used. We will focus in future publications on this topic, however usually 20 points per blade yield sufficiently accurate results for wind farm simulations.

table 3: do you compare with state of the art calculation of NREL5MW actuator line computations? Our EllipSys3D AL implementation is state-of-the-art. The aim of this table is to show the equivalence between the AL without correction and the LL with a viscous core and the AL with the correction and the LL. All existing corrections for the AL rely on tuning some parameters or necessitate highly resolved grids, whereas our correction is purely based on aerodynamic theory. Obtaining a LL solution with an AL is desirable, as the LL method has proven to give accurate results for a large variety of wind turbines.

[Figure]

**Additional changes**

- Doubled the grid resolution to increase the confidence in our results and use the same grid for both length scales to eliminate the possibility of grid effects influencing the results.

- Changed Fig. 2 to display grid convergence in terms of rotor thrust, as now more grid sizes and both length scales are shown.

- Removed $x_\perp$ in Fig. 3

- Fixed Eq. 21, as the definition of $x_\perp$ was incomplete

- Added algorithm overview at the end of Sect. 4.

- Fig. 6, removed some lines to improve readability.

[Figure]

**Bibliography**

Jha, P. K., Churchfield, M. J., Moriarty, P. J., and Schmitz, S.: Guidelines for Volume Force Distributions Within Actuator Line Modeling of Wind Turbines on Large-Eddy Simulation-Type Grids, Journal of Solar Energy Engineering, 136, 031 003, https://doi.org/10.1115/1.4026252, URL http://solarenergyengineering.asmedigitalcollection.asme.org/article.aspx?doi=10.1115/1.4026252, 2014.

---

## Author Response (AR2)

**Review Response**

Dear Editor,

13 May 2019
alrf

Thank you for reviewing the last changes we made to the document. We incorporated your additional comments and improved the line style in figures 6 and 7 and have added a figure of the domain as suggested by one of the reviewers.
We hope the updated document finds your approval.

**Best regards,**

**Alexander Meyer Forsting**
DTU Wind Energy

REG-no. DK 30 06 09 46

**DTU Wind Energy**
Department of Wind Energy

Frederiksborgvej 399
Building 118
4000 Roskilde
Denmark

Tel.. +45 46 77 50 85

www.vindenergi.dtu.dk